# Short-Term Pain Evolution and Treatment Success of Pulpotomy as Irreversible Pulpitis Permanent Treatment: A Non-Randomized Clinical Study

**DOI:** 10.3390/jcm11030787

**Published:** 2022-01-31

**Authors:** Julien Beauquis, Hugo M. Setbon, Charles Dassargues, Pierre Carsin, Sam Aryanpour, Jean-Pierre Van Nieuwenhuysen, Julian G. Leprince

**Affiliations:** 1Adult and Child Dentistry, Cliniques Universitaires Saint-Luc, 1200 Brussels, Belgium; charlesdassargues@hotmail.com (C.D.); pierre.carsin@saintluc.uclouvain.be (P.C.); samaryanpour@gmail.com (S.A.); vannieuwenhuysen@me.com (J.-P.V.N.); 2DRIM Research Group & Advanced Drug Delivery and Biomaterials, Louvain Drug Research Institute, UCLouvain, 1200 Brussels, Belgium; hugosetbon@gmail.com; 3Private Practice, Av. Louise 391, 1050 Brussels, Belgium; 4Private Practice, Rue Edmond Laffineur 9, 1300 Wavre, Belgium; 5Private Practice, All. de la Minerva 2, 1150 Brussels, Belgium; 6Private Practice, Rte du Lion 10, 1420 Braine-l’Alleud, Belgium

**Keywords:** pulpotomy, pulpitis, endodontics, toothache, treatment outcome, tricalcium silicate

## Abstract

The objective of this work was to evaluate (1) the short-term evolution of pain and (2) the treatment success of full pulpotomy as permanent treatment of irreversible pulpitis in mature molars. The study consisted of a non-randomized comparison between a test group (*n* = 44)—full pulpotomy performed by non-specialist junior practitioners, and a control group (*n* = 40)—root canal treatments performed by specialized endodontists. Short-term pain score (Heft–Parker scale) was recorded pre-operatively, then at 24 h and 7 days post-operatively. Three outcomes were considered for treatment success: *radiographic*, *clinical* and *global* success. For short-term evolution of pain, a non-parametric Wilcoxon test was performed (significance level = 0.05). For treatment success, a Pearson Chi square or Fisher test were performed (significance level = 0.017–Bonferroni correction). There was no significant difference between *test* and control groups neither regarding short term evolution of pain at each time point, nor regarding *clinical* (80% and 90%, respectively) or *global* success (77% and 67%, respectively). However, a significant difference in *radiographic* success was observed (94% and 69%, respectively). The present work adds to the existing literature to support that pulpotomy as permanent treatment could be considered as an acceptable and conservative treatment option, potentially applied by a larger population of dentists.

## 1. Introduction

Dental pulp pathologies are often associated with high levels of pain, requiring appropriate local treatment to effectively relieve the patient [1]. While reversible pulpitis is currently managed by vital pulp therapies [2], the treatment of cases diagnosed clinically as irreversible pulpitis is more invasive. It indeed consists in a pulpotomy as emergency procedure [3], followed by complete root canal treatment.

However, a trend towards more conservative strategies has been observed in recent years notably the consideration of pulpotomy as a permanent treatment [4,5,6,7]. This evolution is related to improved knowledge in pulp biology and biomaterials. First, histological studies of teeth with pulpitis have highlighted the existence of a gradient of inflammation within the pulp tissue. Healthy pulp tissue was shown to persist underneath areas with high levels of inflammation and sometimes necrosis [8]. Second, tricalcium-silicate cements were reported in histological studies to have the potential of inducing more favorable pulp responses compared to previous calcium-hydroxide pulp capping materials [9]. A trend in favor of these materials was also identified in clinical studies [9,10].

Preserving pulp tissue by pulpotomies followed by pulp capping has several potential benefits. These include preservation of immunocompetent tissue, reduction in the cost of treatment for both the patient and healthcare systems, and reduction in the treatment complexity and duration. It has nevertheless been stated that additional prospective works are necessary to confirm the positive trend in favor of this treatment strategy [2,11]. Moreover, apart from a few studies [12,13,14], most available works have a rather limited follow-up time (≤12 months).

While some works consider pulpotomy as a treatment of teeth with carious exposures [15,16], regardless of clinical symptoms, others, like the present one, focused on the treatment of clinically-diagnosed irreversible pulpitis. In the latter case, it is important to consider not only long-term outcomes, but also the effectiveness of short-term pain relief. For this purpose, numerical rating scales and/or category judgments that are subsequently converted into numerical values have been widely used as they are easily understood by patients and considered suitable for the measurement of dental pain [17,18,19]. With regard to appropriate times for assessing post-endodontic treatment pain, one day and one week have been described as key time points [20].

Finally, the level of experience of practitioners has, to our knowledge, never been considered in the design of the studies evaluating these procedures. However, as mentioned above, the pulpotomy as permanent treatment is referred to as an easier procedure, therefore potentially accessible to less experienced practitioners, including non-specialists. To take this aspect into account, we designed a non-randomized clinical study to compare the alternative strategy to what is considered as “best available therapy”. Two groups were considered: (1) pulpotomy as permanent treatment performed by non-specialist junior practitioners (test group), and (2) root canal treatment performed by specialized endodontists in private practice (control group).

The objective of the present work was to evaluate (1) the short-term evolution of pain and (2) the treatment success of full pulpotomy as permanent treatment of irreversible pulpitis in mature molars.

## 2. Materials and Methods

### 2.1. Patient Selection

The present prospective study conformed to STROBE guidelines and received approval from the ethics committee of Cliniques universitaires Saint-Luc (Brussels, Belgium) (Reference# 2016/19JAN/016) and was registered at clinicaltrials.gov with registration number NCT02920606.

The study included adult patients with a diagnosis of irreversible pulpitis in mature molar teeth, respecting eligibility criteria for inclusion and exclusion (Table 1). Irreversible pulpitis was defined as spontaneous, radiating pain that lingers after removal of cold stimulus [2]. The inclusion period was from June 2016 to June 2020. An information letter was given to all patients, and informed consent was signed.

In the design of the present study, a difficulty in including patients was expected, due to aspects such as patient motivation (emergency cases) or predictable heterogeneity between groups (population consulting specialists compared to those visiting a dental emergency department). In this context, the present study was initially designed as a pilot study with 50 patients per group as a realistic target.

Systematic forms were used to collect the data for each patient on the day of inclusion.

The study consisted of a non-randomized comparison between two groups (test and control) (Figure 1).

The test group included patients treated in the dental department of Cliniques universitaires St-Luc (Brussels, Belgium). The treatment consisted in a full pulpotomy as permanent treatment, and was performed by non-specialist junior practitioners (≤3 years residents).

The control group included patients treated in private specialist dental practice. The treatment, considered as gold standard, consisted in a root canal treatment performed by specialized endodontists in two different practices.

### 2.2. Clinical Procedure

Following patient inclusion and prior to anesthesia, pain and pre-operative data were collected. Bitewing and a periapical radiographs were taken systematically.

Test group—The tooth was then anesthetized using either Scandonest (3%) or Septanest (4%, 1:200,000 adrenalin) (Septodont, Saint-Maur-des-Fossés, France), respectively, for inferior alveolar nerve block (lower molars) and infiltration anesthesia (upper molars); complementary intra-ligament injections were performed with Scandonest when required. Rubber dam was placed, and the carious lesion was thoroughly excavated when present. The pulp chamber was then accessed with a new sterile bur, and the coronal pulp was completely eliminated. The pulp chamber was rinsed with NaCl, and hemostasis was obtained with a sterile cotton pellet soaked in NaCl. The cavity was gently dried with air spray, and the pulp tissue was capped with a tricalcium-silicate cement (Biodentine, Septodont, Saint-Maur-des-Fossés, France), which was left to set for 15 min. Whenever possible, a permanent composite restoration was placed on the same appointment using a combination of Clearfil SE Bond 2 (Kuraray-Noritake, Japan) and a highly filled composite, either GrandioSO (VOCO, Germany) or Clearfil Majesty Posterior (Kuraray-Noritake, Japan). When permanent restoration could not be placed directly, it was performed within the four weeks after the procedure. All permanent restorations were placed by the investigating team within the dental department. The use of magnification was systematic, mostly loupes, and sometimes microscopes, to properly evaluate pulpal status and compliance with inclusion/exclusion criteria (Table 1).

Control group—No specific recommendations were given on how to perform root canal procedures, considering that the treatments were performed based on the ESE quality guidelines [21]. The aim was to let the specialist practitioners carry out their treatments according to their usual procedure, to be as close as possible to their routine work. The use of operative microscope was systematic, and allowed to evaluate compliance with inclusion/exclusion criteria corresponding to control group (Table 1). They were encouraged to perform their treatments in one appointment whenever possible, but were allowed to delay root canal obturation when time was lacking. A temporary glass ionomer restoration was placed between the appointments. A permanent coronal restoration was placed within the four weeks after the procedure by the referring dentist (either composite, crown or onlay/overlay).

### 2.3. Short-Term Evolution of Pain

Pain score was measured using the Heft–Parker scale [22]. The latter is based on verbal descriptors of clinical pain (used with patients), which are then converted into numerical values (by the investigators) (see Appendix A). The Heft–Parker scale has the specificity of presenting the clinical pain levels with an unequal numerical spacing between pain descriptors, in order to better reflect the differences in word meaning [17].

Besides pre-operative measurement of pain intensity, the investigators conducted a telephone follow-up at 24 h (24 h) and 7 days (7 d) for patients in both groups. The pain scale (verbal descriptors only) was given to patients at the initial appointment to facilitate their evaluation at each timepoint.

In the control group, if the treatment was performed in more than one appointment, pain score was recorded after each appointment and the highest pain level was taken into account for the analysis.

### 2.4. Treatment Success

The evaluation of treatment success was performed in both groups based on similar criteria. One or more controls were performed depending on patient availability and compliance. Bitewing and periapical radiographs were taken systematically at each follow-up appointment. Three outcomes were considered.

*Global* success was defined as the combination of *clinical* and *radiographic* success.

*Clinical* success was defined as the absence of clinical signs (swelling, sinus tract, tooth mobility or deep periodontal probing) and symptoms (pain or discomfort). Therefore, any documented endodontic re-intervention or extraction for endodontic reasons at any time was considered as *clinical* failure.

*Radiographic* success was considered for cases with ≥12 months follow-up. It was determined as follows: first, no root resorption or furcal bone loss shall be observed; second, the periapical index (PAI) evolution shall either be a maintenance or return to healthy status (PAI 1 or 2), a maintenance of PAI 3 or a decrease in PAI if a true periapical lesion was pre-existing (PAI > 3).

The radiographs were submitted to two independent evaluators for analysis (JPVNH and SA). They were previously calibrated on 30 cases selected outside the present study based on the PAI scoring system defined by Ørstavik et al. [23]. The analysis of all radiographs were performed in a random order, under the same light conditions and on the same screen. Since multi-rooted teeth were studied, the highest PAI score at the root level was attributed to the tooth. In case of hesitation between two PAI scores, evaluators were asked to favor the higher one. The evaluation was repeated in a similar manner 3 weeks later. At the end of each session, disagreements were identified by the principal investigators (JB and JGL) and discussed between both evaluators to reach a consensus. Cohen’s kappa coefficient was calculated to assess evaluator agreement.

### 2.5. Histological Analysis

For any successful case extracted for non-endodontic reasons in the *test* group (e.g., non-restorable factures or prosthetic reasons), teeth were placed in paraformaldehyde 4% for fixation and processed for histological evaluation according to the procedure described in [24].

### 2.6. Statistical Analysis

The data were submitted to analysis of normality using the Shapiro–Wilk test.

For short-term evolution of pain, a non-parametric Wilcoxon test was performed to test the effect of group (at each timepoint). For the effect of time, since normality could not be demonstrated, a Wilcoxon signed rank test was used.

For treatment success, a Pearson chi square or Fisher test were performed to compare *clinical*, *radiographic* and *global* success between test and control groups.

A logistic regression or test of independence was performed to test the impact of several variables, on the one hand on attribution of patients to both groups, and on the other hand on treatment success. A significance level of 0.05 was chosen for short-term analyses and sample distribution. Since three outcomes were considered for treatment success, a Bonferroni correction was applied and a significance level of 0.017 was chosen.

## 3. Results

A total of 57 and 41 patients were eligible to participate in the study in the test and control groups respectively (Figure 2), which were subsequently reduced to 44 and 41 patients due to intra-operative exclusion criteria (Table 1).

### 3.1. Short-Term Evolution of Pain

This analysis could be performed for 44 and 40 patients in the test and control groups respectively. There was no significant difference (*p* > 0.05) between both groups at each time point (T0, 24 h, 7 d) (Figure 3). However, a significant reduction in pain was observed between each time point (*p* < 0.0001).

### 3.2. Treatment Success

Follow-up for global success evaluation was possible for 79.5% and 75% of the patients, with a median follow-up of 24 and 20.5 months, and a mean follow-up of 25.9 and 25.6 months, respectively, in test and control groups. The characteristics of the population in both groups are presented in Table 2. The logistic regression (univariate) regarding patient distribution between both groups based on these variables revealed the following significant differences: pre-operative pain to percussion (*p* = 0.049), patient age (*p* = 0.0004) and presence of a carious cavity (*p* < 0.0001).

Five patients experienced *clinical* failure <12 months, four in test group (managed by root canal treatment) and one in control group, which led to tooth extraction due to vertical root fracture.

For patients with follow-up ≥12 months, the evaluation of the radiographs by two independent observers resulted in a consensus PAI scoring, characterized by a Cohen’s kappa coefficient of 0.72 between the two evaluation sessions. The PAI evolution for each patient according to their follow-up duration is available in the Appendix A). Representative examples of the radiographic evolution can be observed in Figure 4a–h for test group and Figure 4i–p for control group.

No significant difference was observed between test and control groups regarding *clinical* (*p* = 0.32) or *global* success (*p* = 0.347). However, a significant difference in *radiographic* success was observed (*p* = 0.014) (Table 3).

None of the variables listed in Table 2 were found to have a significant effect on either *global*, *radiographic* or *clinical* success (univariate analysis; *p* > 0.017).

Among the *clinical* failures in the test group (<12 months, *n* = 4; ≥12 months, *n* = 3), six cases presented evidence of pulp vitality in all canals. A pronounced inflammation was observed, indicated by intense bleeding. One case was re-treated in another practice.

### 3.3. Histological Analysis

One case classified as global success in the test group at 45 months had to be extracted for restorative reasons, i.e., non-restorable fracture (Figure 5a–c). The tooth was processed for histological evaluation (Figure 5d–g), which revealed a vital pulp tissue present in the root canals without any sign of inflammation and with an odontoblastic layer lining the canal walls. It must be noted that difficulties in sectioning were encountered. While demineralized, the collagen remained hard in some areas, preventing serial sectioning throughout the whole sample.

## 4. Discussion

The first major finding was the efficiency of the pulpotomy procedure as permanent treatment in terms of short-term pain relief. A comparable pain reduction at 24 h and 7 d between both treatment procedures was indeed observed. This is in accordance with a randomized clinical study comparing full pulpotomy, selective pulpectomy (restricted to one canal) and full pulpectomy [25]. Another randomized trial reported an even higher and significant pain reduction in the pulpotomy group as compared to the root-canal treatment [26]. The evolution of pain at 24 h and 7 d follows a previously described trend following root-canal treatment, with a moderate drop within one day, and a substantial reduction to minimal levels in 7 days [20]. Hence, pulpotomy as permanent treatment of irreversible pulpitis using tricalcium-silicate cements is at least as efficient as the gold standard procedure in terms of short-term pain relief. Pain relief is an essential component of an endodontic procedure, with pathologies that can lead to very high levels of pain [1,17,18,19], in the same range as those observed in other painful diseases such as renal colic [27,28]. Such levels of dental pain can severely affect patient quality of life during the acute phase, and were, also, shown to be a major cause of acute medical admission following unintentional paracetamol overdose [29,30].

The second major finding was the equivalent efficiency of the pulpotomy procedure as permanent treatment regarding *global* success, as compared to the gold standard procedure. A higher trend was even observed in the test group (77%) but not significantly (*p* > 0.017) compared to the control group (67%). This is in line with a similar observation made in a randomized clinical trial [12]. The latter is currently the most robust study available on the topic, with confirmed irreversible pulp status, >12 months mean follow-up and including a control group.

The incidence (11.4%) of *clinical* failure observed within the first 12 months following the test procedure is in line with the data available in the literature [4,5,6,7]. The overall *clinical* success rate was 80% in our work, all *clinical* failures were related to pain and not to other clinical signs, and lead to re-intervention via standard root-canal treatment. It is important to mention that the location of root canals required the use of magnification, which was possible in all cases. Moreover, the evidence of pulp vitality associated with pronounced inflammation observed in most failed cases connects to the possible interest of quantifying biomarkers such as MMP-9, TNF-α or IL-8 as predictors of the procedure prognosis [4,31,32,33].

The significant difference in *radiographic* success observed in favor of the test group (94% vs. 69%; *p* < 0.017) could be considered unexpected. It cannot be excluded that the higher percentage pre-operative PAI ≥ 3 in the control group affects such finding, but this variable was not identified as statistically significant (*p* > 0.05). Nevertheless, the trend is consistent with the one reported in the literature at two years [13], even considering cases with pre-existing periapical involvement. This trend, illustrated in Figure 4c,d may indicate the positive effect of preserving a living, immunocompetent pulp tissue within the root canal on periapical healing. It is worth noticing that while the presence of a periapical radiolucency (PAI > 3) is not expected in case of an inflamed pulp, some cases were classified as PAI = 4 in the present work and in another [6]. This is due to the fact that the PAI score was not considered in the inclusion/exclusion criterion, which are mainly clinical. Since the radiographic interpretation is performed by independent observers, some cases with PAI > 3 can therefore be reported.

The survival of healthy radicular pulp tissue in successful cases following a pulpotomy as permanent treatment remains unknown. Histology is indeed required to determine the exact status of the pulp tissue, and only few short-term studies provided such information for pulpotomies as permanent treatment in case of irreversible pulpitis. A two-month report on 12 molars reported the presence of healthy radicular pulp devoid of inflammatory signs [34] and a 10-month case report made similar observations in a premolar [35]. In the present work, the histology performed on the successful case provided similar evidence but at 45 months follow-up. In the absence of histology, the observation of a hard tissue barrier underneath the capping material has been considered by some authors as indicator of pulp vitality, as mentioned in a recent review [36]. However, the latter, also, underlines the low reliability of such observation. It is indeed not trivial to determine accurately on a radiograph if an actual barrier has formed, since there is no standardized definition nor are the X-ray images taken in a reproducible manner. Despite the level of evidence regarding the ability of materials to induce mineral bridge formation, it was reported that either pure calcium hydroxide powder or tricalcium-silicate cements are likely the most appropriate materials to cover the pulp [37,38], which was performed here (Biodentine, Septodont, Saint-Maur-des-Fossés, France). The remaining pulp tissue must be free of inflammation, which can be assessed clinically by obtaining a hemostasis. This was a prerequisite in the present work, since it was demonstrated in the context of a direct pulp capping following carious exposure that the degree of pulp bleeding was associated with treatment success [39]. Furthermore, compared to carious exposures, the success rate of pulp capping in traumatic exposures, associated with little or no pulp inflammation, is known to be higher [40]. Nevertheless, it must be noted again that clinical criteria are unfortunately associated with little level of evidence to correlate to histological pulp status [41]. More recently, other intra-operative parameters were subject to discussion, such as the time required to achieve hemostasis [42] or the type and concentration of irrigants [43], without identifying clear trends with regards to treatment outcome. In addition, it was recently shown that in presence of low levels of pulp inflammation (evaluated by MMP-9 quantification), the irrigant (saline vs. 2.5% NaOCl) had little impact on the direct pulp capping outcome [44].

One of the limitations of the study is the modest sample size in each group, but it is relatively in the same range as most available studies on the topic of vital pulp therapy of irreversible pulpitis [4,5,6], except one works with a much larger sample size [12]. In terms of follow-up, apart from the latter providing results at five years, the present study provides the longest duration with an average of 25 months. The recall rate is reasonable given the difficulties associated with this specific topic, although it should ideally be higher given the limited sample size. Emergency patients are indeed quite difficult to follow over time, as they often consult irregularly and/or tend to return to their regular dentist once the emergency is over, both in test and control groups. The number of patients was balanced between test and control groups. Both groups were also shown to be equivalent in terms of initial pain intensity (*p* > 0.05), but not for three other variables. Among these, none had a significant impact on treatment success (*p* > 0.017). However, based on the differences of distribution between both groups, we cannot exclude the existence of a selection bias. The latter, combined with the limited sample size, could account for the lower success rate in the control group. It can also explain the lower *global* success rate in that group compared to those reported in the literature [45,46]. However, these reported success rates shall be compared with those of irreversible pulpitis with caution, as they correspond to cases with “vital pulp”, without further distinction regarding pulpal diagnosis. To our knowledge, the endodontic treatment success rates available regarding irreversible pulpitis are 65.8% at five years [12], which is lower than the rates usually reported for vital pulp cases as a whole.

Another limitation of the work is the restriction to molar cases only. This was determined based on two main criteria: The ease to identify the anatomical transition between coronal and radicular pulp, and the level of treatment complexity. The first criteria was in fact a methodological decision in order to make the procedure more easily reproducible. The second criteria was related to clinical relevance, since both European and American endodontic associations have identified molar teeth as a criteria for treatment complexity [47,48]. Hence, it was the point of the present work to consider the more conservative pulpotomy strategy (test group) as one potentially applied by a larger population of dentists, whereas the gold standard root canal treatment may be restricted to specialized endodontists. It was interesting to assess whether a strict protocol for performing the vital pulp therapy procedure (facilitated by use of magnification) could compensate for the lesser experience of non-specialist practitioners.

Regarding study design, it has been described in the past that while randomized clinical trials are ideal for evaluating the effect of drugs, it is not easily the case for surgical interventions such as endodontic procedures [49]. A double-blind system or the use of a placebo is generally not possible. Moreover, the preference of using “best available therapy” control group rather than a placebo has been described [50] with reference to the World Medical Association Declaration of Helsinki. In the context of irreversible pulpitis, the “best available therapy” would be defined as root canal treatment, but no study to date has considered practitioner experience as an important part of the definition. Both AAE and ESE clearly identify “best available therapy” as one performed by specialized endodontists [47,48]. Hence, it was the purpose of this study to take this aspect into account in the group design, with specialists performing the procedures in the control group, and non-specialists in the test group. In order to be as relevant as possible, specialists working in private practice were selected for control cases, which therefore resulted in a non-randomized study design. While the latter certainly introduces biases on the one hand, it potentially increases the usefulness of the study on the other hand, which is to be taken into account in a study quality aspect [51].

## 5. Conclusions

To conclude, within the limitations of the present work, this prospective non-randomized study tends to support that pulpotomy is an acceptable and more conservative permanent treatment option for irreversible pulpitis in mature molars. Additionally, re-intervention with the gold standard procedure remains possible in case of failure. Hence, despite its non-randomized design and relatively low sample size, the present work adds to the existing literature to support the fact that pulpotomy as permanent treatment could be considered as first line treatment. The latter could be applied by a larger population of dentists, provided that strict inclusion/exclusion criteria and a rigorous clinical protocol are respected. Ideally, the present findings should be confirmed by a multi-centric practice-based study with larger sample size.

## Figures and Tables

**Figure 1 jcm-11-00787-f001:**
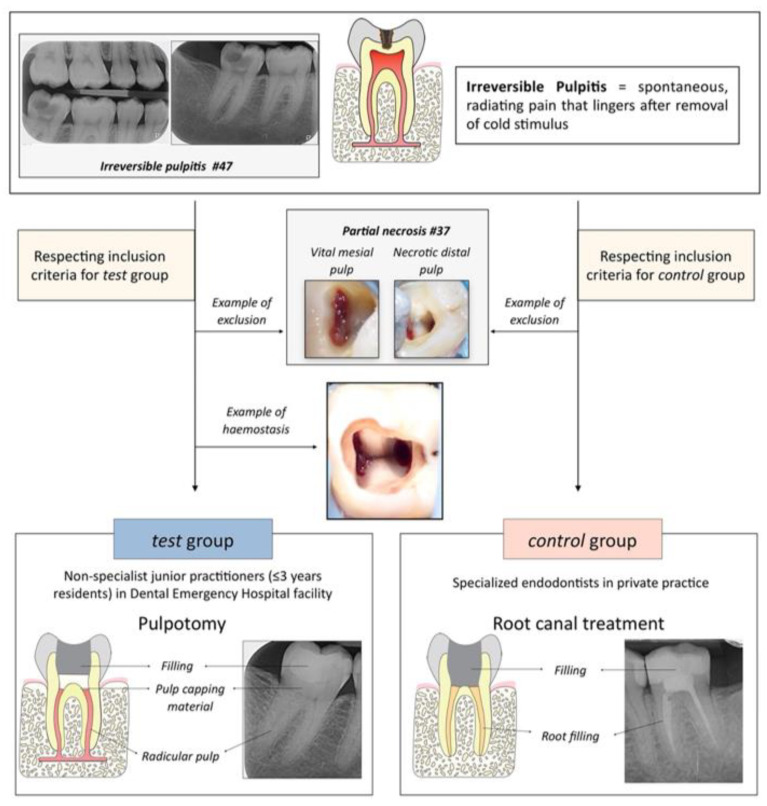
Experimental design.

**Figure 2 jcm-11-00787-f002:**
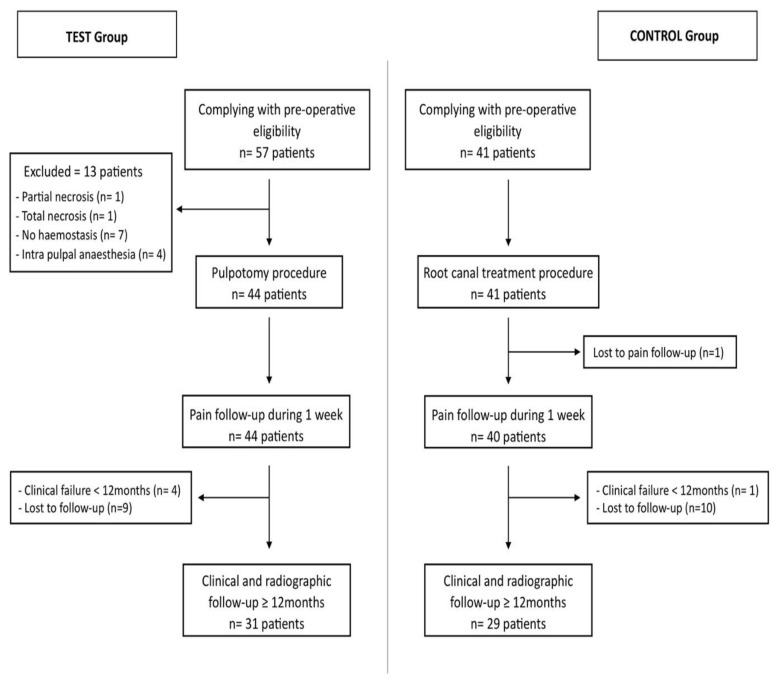
Flowchart for test and control groups based on STROBE recommendations.

**Figure 3 jcm-11-00787-f003:**
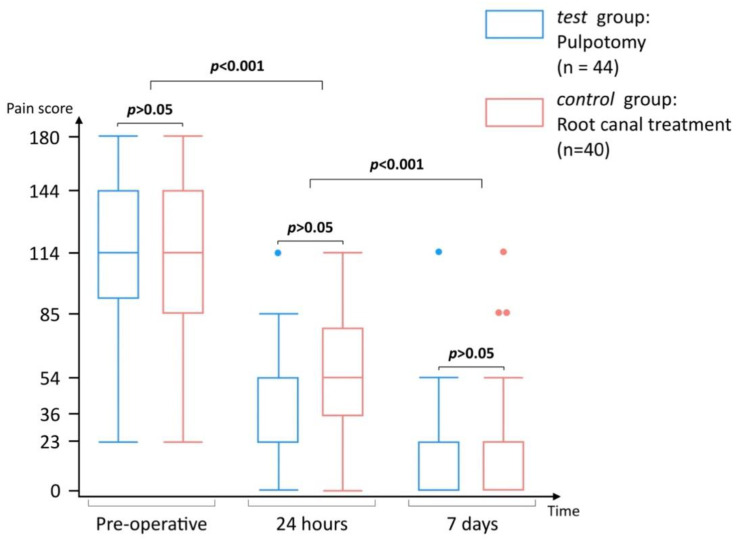
Boxplots of short-term pain evolution for test and control groups; the dots correspond to outliers.

**Figure 4 jcm-11-00787-f004:**
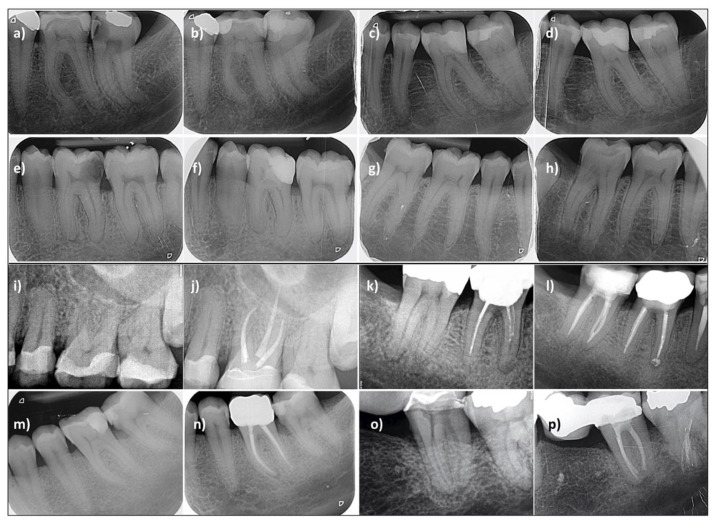
(**a**–**h**) test group—4 clinical cases of pulpotomy. (**a**,**b**) #37 Healthy status quo, follow-up at 46 months; (**c**,**d**) #36 peri-apical healing, follow-up at 20 months; (**e**,**f**) #36 peri-apical healing, follow-up at 16 months; (**g**–**h**) #47 peri-apical aggravation, follow-up at 20 months. (**i**–**p**) control group—4 clinical cases of root canal treatment. (**i**,**j**) #26 Healthy status quo, follow-up at 47 months; (**k**,**l**) #47 peri-apical status quo, follow-up at 50 months; (**m**,**n**) #36 peri-apical aggravation, follow-up at 24 months; (**o**,**p**) #37 peri-apical healing, follow-up at 12 months.

**Figure 5 jcm-11-00787-f005:**
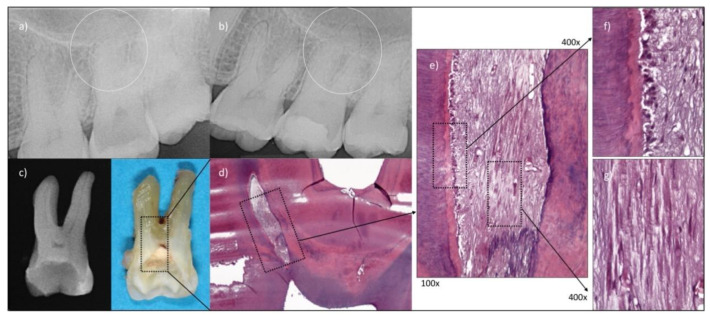
Histological analysis of a case considered as global success, extracted for restorative reason, performed by Domenico Ricucci. (**a**) #27 pre-operative situation, PAI score:1; (**b**) #27 healthy status quo, follow-up at 45 months; (**c**) radiographic and clinical images of extracted tooth; (**d**–**g**) histological sections (H&E staining) at various magnifications, revealing the presence of a vital pulp tissue in the root canals without any sign of inflammation or tissue disruption. An odontoblastic layer lining the canal walls can be observed.

**Table 1 jcm-11-00787-t001:** Eligibility criteria for inclusion in the clinical study.

Pre-Operative Criteria:	Intra-Operative Criteria:
INCLUSION - Adult patient (≥ 18 years) - Molar teeth - Irreversible pulpitis diagnosis (spontaneous pain, radiating pain that lingers after removal of cold stimulus) EXCLUSION - Internal/External resorption - Root and crown fracture - Presence of a sinus tract - Anormal mobility - Immature apex - Swelling - Systemic condition - Non-independent patient - Non-restorable tooth - Periodontal pocket ≥ 6mm - Participation in other medical studies	EXCLUSION - Pulp necrosis in at least one root canal (both groups) - Need of intra-pulpal injection (test group only) - Impossibility to achieve pulp hemostasis at the root canal entrance (test group only)

**Table 2 jcm-11-00787-t002:** Characteristics of cases included in treatment success evaluation (*N* = 65).

Variables	Total *n* (%)	Test Group (Pulpotomy) *n* (%)	Control Group (Root Canal Treatment) *n* (%)
Patients (*n*)	65 (100)	35 (100)	30 (100)
Age (years)			
Mean	39.4	34.8	46.7
Median	38.5	29	49
Male	22 (33.8)	9 (25.7)	13 (43.3)
Dental arch			
Maxillary teeth			
First molar	8 (12.3)	5 (14.3)	3 (10)
Second molar	10 (15.4)	4 (11.4)	6 (20)
Mandibular teeth			
First molar	17 (26.2)	11 (31.4)	6 (20)
Second molar	29 (44.6)	14 (40)	15 (50)
Third molar	1 (1.5)	1 (2.9)	0 (0)
Pre-operative masticatory pain	38 (58.5)	18 (51.4)	20 (66.7)
Pre-operative pulsatile pain	40 (61.5)	18 (51.4)	22 (73.3)
Pre-operative percussion pain	37 (57)	16 (45.7)	21 (70)
Duration of pre-operative pain			
≤2 days	19 (29.2)	13 (37.2)	6 (20)
>2 days–≤ 7 days	15 (23.1)	6 (17.1)	9 (30)
>7 days–≤14 days	5 (7.7)	0 (0)	5 (16.7)
>14 days	26 (40)	16 (45.7)	10 (33.3)
Pre-operative PAI			
1	30 (46.2)	18 (51.5)	12 (40)
2	22 (33.8)	13 (37.1)	9 (30)
3	10 (14.4)	4 (11.4)	6 (20)
4	3 (4.6)	/	3 (10)
Presence of carious cavity	25 (38.5)	21 (63.4)	4 (13.3)

**Table 3 jcm-11-00787-t003:** Characteristics of cases included in treatment success evaluation (*N* = 65).

Variables	Test Group (Pulpotomy) % (*n*)	Control Group (Root Canal Treatment) % (*n*)
*Clinical* success	80 (28/35)	90 (27/30)
*Radiographic* success *	94 ** (29/31)	69 ** (20/29)
*Global* success	77 (27/35)	67 (20/30)

* excluding clinical failures <12 months. ** Significant difference (*p <* 0.017).

## Data Availability

The data presented in this study are available on request from the corresponding authors.

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
