# Peer review of "Short-Term Pain Evolution and Treatment Success of Pulpotomy as Irreversible Pulpitis Permanent Treatment: A Non-Randomized Clinical Study"

_jcm, 2022, doi:10.3390/jcm11030787_

Round 1
Reviewer 1 Report
Dear Authors,
MeSH terms should be used for the keywords section. Note that "vital pulp therapy" is not a MeSH term and should be corrected.
Pain assessment and repetitive scaling as subjective parameters should be considered in the introduction.
The use of diagrams is advantageous in presenting the methodology, but figure 1 is too complex. The lines 130-131 should be clarified, as well as the single time T0 of the scale presentation.
What are the values of the inter and intraobserver error?
Line 165-166 should be clarified, namely "non-endodontic reasons."
The discussion section should be improved. The use of quality of life assessment should be discussed, including pain as a component. Discussion of "pain" as a value concerning age and sex should be considered, which implies characterizing the sample at this level. The categorization of "low" (line 265 and others) and "significant" (line 274) must be referenced and explained, and we cannot use these terms throughout the discussion regularly.
The conclusions are too ambitious, it should be considered according to the study.
Reviewer 2 Report
I want to thank the Editor and the Authors for considering me for the review of this interesting paper which describes a non-randomized clinical trial comparing full pulpotomy Vs root canal treatment for molar permanent teeth with irreversible pulpitis.
The topic is quite new and, in the literature, there is still not sufficient evidence to support full pulpotomy as a definitive treatment for irreversible pulpitis, so new studies are encouraged.
The methods and the results are clear and well described.
In my opinion, the weak points of the paper are:
- no randomization (already addressed by authors)
- high % lost patients: this aspect should be better addressed in the discussion
- the global success in the control group is frankly lower compared to the success of root canal treatment in vital teeth as reported by systematic reviews (e.g. Ng et al. https://doi.org/10.1111/j.1365-2591.2007.01323.x). This point should be better discussed
- which is the meaning of blu and red circles in figure 3, over the 24h and 7 days diagrams?
- sometimes vital teeth with inflamed pulps could show periapical lesions on Rx, but usually there are not periapical radiolucencies. This aspect should be mentioned with reference to the 3 cases with PAI=4.
I think that the most important result of the paper is not the apparent superiority of full pulpotomy over root canal treatment, since the comparison of the two treatments is weakened by many biases, but the overall good clinical outcome of the full pulpotomy group.
